# Primary Sclerosing Cholangitis-Associated Cholangiocarcinoma: From Pathogenesis to Diagnostic and Surveillance Strategies

**DOI:** 10.3390/cancers15204947

**Published:** 2023-10-11

**Authors:** Elisa Catanzaro, Enrico Gringeri, Patrizia Burra, Martina Gambato

**Affiliations:** 1Gastroenterology, Department of Surgery, Oncology, and Gastroenterology, Padova University Hospital, 35128 Padova, Italy; 2Multivisceral Transplant Unit, Department of Surgery, Oncology, and Gastroenterology, Padova University Hospital, 35128 Padova, Italy; 3Hepatobiliary Surgery and Liver Transplantation Center, Department of Surgery, Oncology, and Gastroenterology, Padova University Hospital, 35128 Padova, Italy

**Keywords:** cholangiocarcinoma, primary sclerosing cholangitis, pathogenesis, diagnosis, surveillance, biomarkers

## Abstract

**Simple Summary:**

Cholangiocarcinoma is a major concern in patients with primary sclerosing cholangitis, and it is associated with a high mortality rate. Recent data have demonstrated several differences between primary sclerosing cholangitis-associated cholangiocarcinoma and de novo cholangiocarcinoma in terms of epidemiology and pathogenesis, raising several questions about the pathological mechanisms and the best clinical approach to this tumor. We have reviewed the major updates in primary sclerosing cholangitis-associated cholangiocarcinoma to highlight the possible future diagnostic and surveillance strategies.

**Abstract:**

Cholangiocarcinoma (CCA) is the most common malignancy in patients with primary sclerosing cholangitis (PSC), accounting for 2–8% of cases and being the leading cause of death in these patients. The majority of PSC-associated CCAs (PSC-CCA) develop within the first few years after PSC diagnosis. Older age and male sex, as well as concomitant inflammatory bowel disease (IBD) or high-grade biliary stenosis, are some of the most relevant risk factors. A complex combination of molecular mechanisms involving inflammatory pathways, direct cytopathic damage, and epigenetic and genetic alterations are involved in cholangiocytes carcinogenesis. The insidious clinical presentation makes early detection difficult, and the integration of biochemical, radiological, and histological features does not always lead to a definitive diagnosis of PSC-CCA. Surveillance is mandatory, but current guideline strategies failed to improve early detection and consequently a higher patient survival rate. MicroRNAs (miRNAs), gene methylation, proteomic and metabolomic profile, and extracellular vesicle components are some of the novel biomarkers recently applied in PSC-CCA detection with promising results. The integration of these new molecular approaches in PSC diagnosis and monitoring could contribute to new diagnostic and surveillance strategies.

## 1. Introduction

Cholangiocarcinoma (CCA) has always represented a challenging tumor with poor diagnostic and therapeutic tools and associated very low patient survival. Currently, CCA represents around 15% of all primary liver cancers, with an increasing trend during the last decades due to better awareness of the disease and diagnostic advances, accounting for 3% of all gastrointestinal neoplasms [1,2,3]. CCA is a part of a heterogeneous group of tumors, arising from any tract of the biliary tree, characterized by various genetic mutations, histological subtypes, and clinical presentations. Many risk factors could contribute to CCA development, and some of them are strongly associated with specific geographic areas. Flukes infections (Opisthorchis viverrini and Clonorchis sinensis) represent a major risk factors in the South Asian countries, explaining the higher incidence of CCA in this geographical area (6/100,000 cases per year) [1,3,4]. Incidence in western countries is significantly lower (<4/100,000 cases), and CCA is associated with some risk factors such as viral hepatitis (HBV, HCV) and metabolic dysfunction-associated steatotic liver disease (MASLD), but mostly the presence of an underlying biliary disease. Primary sclerosing cholangitis (PSC) represents the main risk factor for CCA in the western countries, followed by other conditions such as congenital bile-duct cysts, Caroli disease, or choledocholithiasis [5,6,7]. PSC is a rare cholestatic disease affecting the biliary tract, with formation of fibrotic biliary strictures and consequent dilatations along the biliary tree [8]. The high risk of developing hepatobiliary malignancies in PSC is well known and also involves other type of neoplasms, such as gallbladder carcinoma (GBC) and hepatocellular carcinoma (HCC). Moreover, the association with inflammatory bowel diseases (IBD) in up to 80% of PSC patients increases their oncological risk, as they are more prone in developing colorectal carcinoma (CRC). Regarding PSC-CCA, the pathogenetic mechanisms are not fully understood; immune-mediated hits and bile stasis might lead to chronic inflammation and cancerogenesis [9]. PSC-associated CCA (PSC-CCA) is characterized by peculiar epidemiological, clinical, molecular, and genetic features, arising the suspicion of PSC-CCA being a distinct disease from de novo CCA.

## 2. Epidemiology

Despite PSC-CCA representing a small part of all biliary neoplasms, reaching an average frequency of almost 10%, CCA is the most frequent tumor in patients with PSC [10] Indeed, PSC is associated with 400–600-fold higher risk of developing CCA compared to the general population [10,11]. The assessed annual risk ranges between 0.5% and 1.5% [12,13]. According to that, PSC patients are affected by 5- and 10-year cumulative risk of developing CCA around 7% and 6–11%, respectively [1,14]. The reported lifetime prevalence of PSC-CCA ranges from 6 and 13% [11,15,16,17,18]. The risk of developing CCA tends to be higher soon after PSC diagnosis; in fact, approximately 30–50% of CCA are diagnosed within the first year [11,17,19,20]. Despite the significative two-fold reduction of mortality in PSC patients who undergo scheduled imaging surveillance [21], CCA still accounts for almost 30% of all PSC-related deaths, being the main cause of death in these patients [11,12,17,22,23]. Furthermore, among those affected by PSC-CCA, up to 80% of patients die within the first year after diagnosis [24]. The frequency of CCA in patients with PSC is highly variable, as past studies reported a wide range between 4% and 36% [17,25,26,27,28]. The variability is mainly due to the retrospective and monocentric design of the studies, including a limited number of patients, and to different patient populations with heterogeneous tumor characteristics. There is a higher risk of PSC-CCA in Northern Europe and North America than in Asia [29] and Southern Europe [30,31] where PSC tends to have a more benign course and a lower risk of neoplastic complications. More recent studies (on larger and multicentric populations) reported a higher frequency of CCA in PSC, registering a range from 2% to 8% [1,12,19,32,33]. In transplant centers, the frequency of PSC-CCA seems to be higher (up to 19.9%), probably associated with a higher prevalence of patients with advanced PSC [28]. Patients with PSC are more prone to develop perihilar CCA (pCCA) with an odds ratio of 453 (95% CI: 104–999), but there is a higher risk of developing also intrahepatic CCA (iCCA) and distal CCA (dCCA) with odds ratios of 93.4 (95% CI: 27.1–322) and 34.0 (95% CI: 3.6–323), respectively [34]. The clinical presentation of PSC-CCA is difficult to recognize also because the clinical and radiological features are overlapping in the two diseases, and CCA could appear as a precipitating event in misdiagnosed PSC [8,35]. However, the majority of CCA develops asymptomatically, representing an incidental event in up to 40% of all PSC-CCA, also discovered in liver explant pathology or at autopsy [18]. PSC-CCA usually occurs at younger ages, around the fourth decade, while de novo CCA rarely develops before the seventh decade [9,17,36,37].

## 3. Risk Factors

Among the risk factors for PSC-CCA development, age has an important role. Advanced age has been observed as a risk factor for PSC-CCA development in many studies [18,38,39]. Furthermore, the risk increases in each decade ranging from 1.2 to 21.0 per 100 patient-years within the second and seventh decade [12]. The risk is also increased in those with an older age at diagnosis [40,41]. Despite that, the first 1–2 years after PSC diagnosis seems to be a crucial timepoint, due to a higher risk of developing PSC-CCA, regardless of the age. Children have a lower risk of developing CCA, eventually due to the increased prevalence of PSC/autoimmune hepatitis (AIH) variant, accounting for a lower rate of PSC-CCA [42]. Small-duct PSC (sdPSC) has also a lower risk of CCA development, possibly due to a different pathogenetic mechanism not involving large ducts, normally affected by the malign transformation [43,44,45]. Male sex is another risk factor for PSC-CCA [12,17], possibly due to the higher risk of association with environmental factors that predispose to neoplasms (smoke, alcohol, etc.) [25]. Presence of IBD [18,38,39], as well as the prolonged duration of IBD in PSC patients [46], increases the risk of PSC-CCA development. The association between IBD and PSC-CCA has been confirmed by the International PSC Study Group (IPSCSG), including 7121 patients with PSC. They showed a higher risk of hepatobiliary malignancy in patients with IBD, particularly ulcerative colitis (UC), compared to those without IBD. Regarding the presence of biliary stenosis, “dominant strictures”, called high-grade strictures in the recent guidelines [47], have been associated with an increased risk of biliary neoplasms [48], and the estimated risk of any stricture to progress to CCA is between 5% to 26% [49,50,51]. However, the presence of high-grade strictures does not represent biliary malignancy, and up to 50% of PSC patients develop benign stenoses during the course of the disease [52]. On the other hand, CCA might also develop in absence of high-grade strictures; therefore, the real impact of the presence of high-grade strictures on tumor development is still unknown. Unexpectedly, the duration and severity of PSC do not seem to correlate with PSC-CCA development [20,38], as well as the presence of cirrhosis; in fact, the majority of PSC-CCA develop on a non-cirrhotic liver [17,24].

## 4. Pathogenesis

The most supported model of biliary carcinogenesis is the “multistep carcinogenesis” model. Presence of chronic inflammation leads to the sequential progression from damaged normal biliary epithelium to low-grade dysplasia (LGD), high-grade dysplasia (HGD), and finally invasive cancer [53,54,55]. A higher frequency of metaplasia and dysplasia was observed in the explanted liver of patients who developed PSC-CCA than PSC alone, suggesting that these histological findings could anticipate CCA development [43,53,54]. Not all types of metaplasia have the same transformative potential, as intestinal metaplasia seems to be more typical of PSC-CCA in comparison with other forms of metaplasia (mucinous, pyloric, etc.) that equally develop in both presence and absence of an underlying PSC. Intestinal metaplasia represents a significant predictor of both dysplasia and CCA, confirming the existence of a multistep process of neoplastic transformation and partially differentiating the pathogenesis of PSC-CCA from CCA alone. This observation could explain the different epidemiology PSC-CCA compared with de novo CCA and the earlier development of PSC-CCA [53]. Nevertheless, the role of inflammation in PSC and its progression to neoplastic transformation is still unknown, as well as the pathogenesis of the disease itself. Inflammation in PSC develops on a complicated interplay between immune-mediated mechanisms and external biliary insults from exogenous antigens (infectious, bile etc.), as shown in Figure 1.

The gut microbiota is believed to play an important role in the pathogenesis of PSC, also supported by the strong association between PSC and IBD [56,57]. The presence of a “leaky” intestinal mucosa, especially when there is a concomitant active IBD, seems to promote translocation of gut microbes and activated gut-specific lymphocytes through the portal vein system to the liver, generating a local inflammatory response in the biliary ducts. Furthermore, the intestinal, as well as biliary, microbiota composition in both PSC and PSC-IBD shows unique features, with an abundance of certain bacteria genera such as *Veillonella*, *Enterococcus*, and *Streptococcus*. Still, variations in fungal and viral species are also noticeable, and there is a generalized reduction in short-chain fatty acid metabolisms, which play a pivotal role in maintaining mucosal integrity, and a decrease in microbiota diversity [58]. Moreover, prior research showed both intestinal and biliary microbiota-specific changes compared to healthy controls [59,60]. However, the precise implications of these alterations remain largely unexplored, especially in the contest of PSC-CCA. Subsequent investigations are warranted to comprehensively characterize and elucidate the functional significance of these microbial alterations and how they could contribute to the transformation of inflammatory changes of the biliary tree to cancer.

Bile toxicity, related with susceptibility genes involved in bile acid homeostasis like TGR5 or CTFR [61] and the impaired production of the HCO3− biliary umbrella [62], seems also to participate to biliary damage. The bile of PSC and PSC-CCA patients also presents reduced levels of phosphatidylcholine, an essential component of the mixed micelles that protect cholangiocytes from the detergent properties of bile acids [63]. Moreover, PSC is strongly associated with some human leukocyte antigen (HLA) class I/class II haplotypes, sharing a “common ground” with several autoimmune diseases [64], and other susceptibility genes participating in affecting mechanisms of inflammation and apoptosis (NFKB1) dysregulating cytokinin pathways (IL2, IL2RA, IL21) and generating an aberrant antigen presentation and lymphocytes activation (TNFRSF14, CTLA4) [65,66,67].

The external triggers (bacteria translocation, bile toxicity, etc.) are likely to activate this aberrant adaptive immune system characterizing PSC, stimulating the transition into an activated phenotype of biliary epithelium [68]. In chronic inflammation, cholangiocytes manifest a high proliferative capacity as an adaptative response to prevent ductopenia, and could also act as facultative liver stem cells in case of important liver damage [69]. Inflammation also generates the expansion of the peribiliary gland compartment activating proliferation and hyperplasia of the biliary tree stem cells contained in the peribiliary niche [70,71]. Cholangiocytes proliferation is extremally heterogeneous, involving two differentiated phenotypes [72]: (1) the ductular reactive cholangiocytes, which seem to derive from the expansion of proliferative cholangiocytes and the peribiliary gland stem cells [73,74] as described above (less likely, they could also derive from trans-differentiated hepatocytes) [75]. This transition into reactive cholangiocytes has a crucial role in developing peribiliary fibrosis through the crosstalk with myofibroblasts, along the Hedgehog pathway [70] and the hepatic stellar cells activation [76], stimulating the excessive extracellular matrix deposition and fibrotic stricture development [77]. (2) The senescent cholangiocytes, deriving from an extended DNA damage primarily involving N-Ras activation (with senescence-associated markers like p16(INK4a) and γH2A.x) [78]. These cells share a senescence-associated secretory phenotype (SASP) [79], with aberrant oncogenes and transcription factors (like ETS1) expression [80] that promote apoptosis resistance, and secretion of several cytokines (i.e., IL-6, IL-8, CCL2, PAI-1) associated with the development of CCA [78]. Previous studies have already described the presence of p53 dysfunction as the most common aberration in PSC-CCA, usually occurring in up to 30% of cases, more frequently than sporadic CCA [81]. Also, presence of *ERBB2* amplification and *SMAD4* loss are more frequent in PSC-CCA [82,83]. Other common mutations involve *KRAS*, *GNAS*, and *PIK3CA* [82], as well as loss of chromosome 9p21 and inactivation of p16 [28]. Interestingly, some alterations like copy number variation, an indicator of chromosomal imbalance, occur earlier in PSC-CCA cancerogenic cascade [82].

Among cytokines, IL-6 seems to have a mitogenic potential in CCA, promoting the expression on myeloid cell leukemia 1(McL-1) through STAT3 pathway, and concomitant inhibition of SOCS3 [84,85]. IL-6 also promotes PI3K/AKT pathway, resulting in antiapoptotic stimuli [86]. An important role in cellular damage is played by biliary infections, promoting oxidative stress of cholangiocytes. Biliary infections activate the inducible nitric oxide synthase (iNOS) in inflamed tissue, with a consequent overproduction of nitric oxide (NO) and other reactive oxidants (e.g., 3-nitrotyrosine), which promotes direct DNA damage and inhibition of base excision repair [87,88]. During infections, bacteria-derived lipopolysaccharide (LPS) also has a role in activating the epidermal growth factor receptor (EGFR) and the cyclooxygenase-2 (COX-2) and consequent stimulation of bile duct proliferation [89], as well as the accumulation of some bile acids like deoxycholic acid, which could promote several mechanisms such as enhanced expression of interleukin IL-6 and downregulation of farnesoid X receptor (FXR)-dependent chemoprotection, which could facilitate cancer development [90]. Genetic and epigenetic factors could contribute to PSC-CCA developments. For instance, natural killer cell receptor G2D (NKG2D) normally interacts with its ligand, major histocompatibility complex class I chain-related molecule A (MICA), with a protective role against tumor development. However, some polymorphisms of NKG2D seem to be associated with a lower receptor–ligand affinity and higher risk of PSC-CCA development [91]. Among epigenetics changes, methylation of 9p21was seen to be a possible strategy of p16 silencing [28]. Regarding microRNAs (miRNAs), their expression was seen to be associated with PSC and its progression to CCA. First observations came from mice models of chronic cholestasis. Induction of cholestasis was associated with acceleration of CCA development, which came with miRNAs down-regulation (miR-34a) or up-regulation (miR-210) [92]. Voigtländer et al. found a significative expression of some miRNAs in the bile of PSC-CCA patients, most of them involved in the processes of epithelial–mesenchymal transition [93]. Those miRNAs’ expression differs significantly in PSC and PSC-CCA, with higher levels in the second group. Interestingly, the levels of this specific miRNA were substantially lower in patients with CCA alone, suggesting a different etiopathogenesis in CCA developing on an underlying biliary disease.

## 5. Clinical Presentation

Most CCAs evolve asymptomatically in the early stages, and it is often difficult to clinically distinguish between CCA complicating a PSC and PSC alone. Although some studies proved the absence of symptoms to be a favorable prognostic factor in PSC-CCA patients [25], the insidious nature of this neoplasia could be one of the main factors of late diagnosis and poor prognosis. When symptomatic, CCA could present (and should be always suspected) with a rapid change in clinical condition, usually with a decline in liver function and worsening of jaundice, associated with weight loss, asthenia, and abdominal pain. Nevertheless, these are non-specific symptoms that also occur in benign complications of PSC. Symptoms of CCA also depend on the site of the malignancy development. In general, the most common form is the eCCA, mainly represented by pCCA (almost 50–65% of PSC-CCA cases), followed by dCCA (20–40%). Finally, iCCA is the least represented form (15%) [94,95]. Clinical presentation of extrahepatic dCCA or pCCA is characterized by severe cholestasis due to biliary obstruction, with appearance of jaundice, pruritus, and malaise, associated with alteration of liver biochemistry tests, like alkaline phosphatases (ALP) and bilirubin. According to that, increasing levels of bilirubin in patients with PSC should always be investigated. On the other hand, iCCA tends to be completely asymptomatic or with mild and non-specific symptoms such as abdominal pain, reduced appetite, and malaise. Biochemical alterations only appear in advance stages without significant signs of biliary obstruction [96]. The iCCA form represents the most insidious with a high risk of late diagnosis.

## 6. Diagnosis

Despite the improvement in PSC-CCA detection during the years, early diagnosis is still difficult to achieve, and there is no test that could detect and stage the tumor at the same time [97]. According to that, multiple modalities including biochemistry, imaging, and biliary invasive techniques need to be integrated to guide toward diagnosis of malignancy. Principal guidelines suggest investigating on any sudden clinical or biochemical alteration, as well as radiological progression or new diagnosis of high-grade stenoses, regardless the symptoms [47,98]. Non-invasive imaging techniques represent a useful and simple diagnostic tool for PSC-CCA detection. Ultrasound (US) computed tomography (CT)/contrast-enhanced CT scans and magnetic resonance imaging (MRI) with magnetic resonance cholangiopancreatography (MRCP) represent the standard diagnostic imaging used. Among them, MRI/MRCP is characterized by a better diagnostic performance than CT in terms of contrast resolution and enhancement. On the other hand, CT is preferable for staging CCA, due to its better capacity to detect vascular involvement and extrahepatic metastases, with an accuracy of over 85% [99,100]. However, despite an almost 100% of specificity of all those imaging techniques, sensibility remains scarce around 32% for MRI/MRCP, and even lower for CT and US (25% and 10%, respectively) [101]. Moreover, there are no universally accepted standards for MRI features. The MRI working group of the IPSCSG provided a workup for PSC diagnosis, useful also for the diagnosis of malignancies, including a T2-weighted (T2w) 3D or 2D MRCP for the correct visualization of the biliary tree, and the use of both T2w axial and coronal sequences and T1-weighted (T1w) both pre-contrast and contrast-enhanced, preferably using fat suppression. The use of diffusion-weighted imaging (DWI) could enhance the capacity of distinguishing malignancies from benign lesions, as well as magnetic resonance elastography, when available [102]. Regarding MRCP quality, an accurate PSC evaluation would depict up to third-order biliary ducts, without any artifact or blurring of the ducts along the biliary tree [103]. The use of standardized criteria and the referral of PSC patients to dedicated radiologists in the field might be potential approaches to improve radiological sensibility in PSC-CCA detection.

The use of F18-fluorodeoxyglucose positron emission tomography (FDG18-PET) has not been extensively studied in patients with PSC, but previous experiences showed a low specificity for sporadic CCA diagnosis [104,105]. The limited experience from Fevery et al. including 10 PSC patients underwent FDG18-PET during LT evaluation showed high-sensitivity lesion detection; however, three patients were discovered as false positives on the explant pathology [106]. FDG18-PET in PSC-CCA diagnosis is affected with a high rate of false positives, mainly due to the inability to differentiate inflammatory strictures from malignant lesions. According to that, current guidelines do not recommend the use of PET for PSC-CCA diagnosis [47]. Among biomarkers, carbohydrate antigen 19-9 (CA 19-9) represents the most used to guide diagnosis of PSC-CCA and for surveillance; however, its role remains still largely limited. Although CA 19-9 increase was seen to be associated with the neoplasm development [107], is not rare to find PSC-CCA developing with a negative or scarce increase of CA 19-9 [108]. The CA 19-9 expression showed an interpersonal variability depending on fucosyltransferase 2 and 3 (FUT2-3) polymorphisms, especially in Caucasian ethnicity, with inadequate synthesis of the biomarker and increasing of false negative results [109]. Many CA 19-9 cut-offs have been investigated to establish the optimal value in PSC-CCA diagnosis. Using the cut-off value of 129 U/mL, a first study from Levy et al. provided a high specificity (98.5%) and a good sensitivity (78.6%), with an adjusted positive predictive value of 56.6%. Unfortunately, only a few patients with detected PSC-CCA were candidates for curative therapy [108]. A subsequent Sweden study using the same cut-off of 129 U/mL or higher showed a critically lower sensitivity of 13% in patients with PSC [101], and data from the Mayo Clinic showed that more than one-third of patients reported with CA 19-9 levels above 129 U/L did not actually have CCA [110]. The retrospective analysis from Charatcharoenwitthaya et al., cited above, investigated several cut-off values, observing that the value with the highest diagnostic yield was 20 U/mL, achieving 78% and 67% of sensitivity and specificity, as well as a positive predictive value of 23% and negative predictive value of 96% [101]. Furthermore, the concomitant use of an imaging technique could enhance sensitivity up to 91% with CA 19-9 plus US and 100% with CA 19-9 plus MRI/MRCP [101], reaching an extremally high negative predictive value that could rule out PSC-CCA in presence of both serological and radiological negativity. However, it highly affected specificity with a higher risk of false positives [111]. Furthermore, CA 19-9 fluctuations are strongly influenced by many other conditions, such as other malignancies like pancreatic tumors, associated with a higher incidence in PSC patients than the general population [17,18,112], or benign diseases such as superimposed bacterial of fungal cholangitis [113,114]. Carcinoembryonic antigen (CEA) has been proposed as a biomarker for early detection of PSC-CCA, being less influenced by bacterial cholangitis [115]. However, it remains inferior to CA 19-9 [116] with a sensitivity of 33–68% and specificity of 82–89% for values superior to 5 ng/mL [117]. The sensitivity seems to be higher with the joint use of both CA 19-9 and CEA, showing diagnostic yield of 86% [118], but with an increase in false positive results [117].

Despite the strong effort in searching for non-invasive diagnostic biomarkers, tissue sampling remains a diagnostic tool for the diagnosis of many PSC-CCA. Cholangiographic techniques, mainly endoscopic retrograde cholangiopancreatography (ERCP), represent the best approach in malignancies localized in the extrahepatic biliary tract, which accounts for the majority of CCA developing on PSC [119]. Percutaneous transhepatic cholangiography (PTC) is becoming more obsolete due to the higher difficulty of execution and risk of complication [120]. Currently, the use of ERCP is subsequent to MRI/MRCP due to similar sensitivity [121] and more benefit in terms of cost-effectiveness [122] and remains necessary when non-invasive imaging techniques are not feasible or able to provide a high suspicion of PSC-CCA that justifies therapeutical intervention (resection, LT) without histologic or cytologic confirmation. Furthermore, the use of ERCP is extremally useful in presence of relevant strictures (high-grade strictures with symptoms or signs of biliary obstruction according to more recent guidelines [47]) in order to combine both diagnostic sampling and therapeutical intervention [48]. However, endoscopic sampling via endoscopic retrograde cholangiopancreatography (ERCP), despite being highly specific, is limited by poor sensitivity. In fact, brushing cytology, according to the literature including a systematic review and meta-analysis, provides only a 43% sensitivity [123,124]. The use of repeated brushing passes during the same cytologic sampling showed only a scarce improvement of sensitivity, up to 57% [125]. The use of fluorescence in situ hybridization (FISH) can enhance sensitivity and improve the diagnostic yield of brush cytology. The sequential use of FISH after biliary cytology in equivocal cases, evaluated by the retrospective study from von Seth et al., provided a significant improvement in sensitivity, up to 80% (95% CI: 52–96), with a constantly high specificity of 96% (95% CI: 92–98) and a predictive positive and negative value (PPV and NPV) of 60% (95%CI: 36–81) and 98% (95% CI: 95–100), respectively [126]. The digital image analysis (DIA), quantifying cellular DNA content and other nuclear features to distinguish the nature of biliary stenoses, did not become popular due to the requirement of a highly cellular sample and a reduced specificity than brushing cytology [127]. The combination of brushing with endoscopic retrograde trans-papillary forceps biopsy demonstrated an improved diagnostic performance compared with brushing cytology [128] but reaching only a little advantage in sensitivity when the two techniques were combined (up to 60%) [129]. Better results were derived from the association with other techniques like FISH or bile aspiration cytology [130]. The triple modality testing, including brush cytology, biopsy, and FISH, is able to enhance sensitivity from 42% of cytology alone to 82%, with similar specificity (100%). Therefore, according to the European Society of Gastrointestinal Endoscopy (ESGE) and EASL, forceps biopsies should be considered in all patients with suspicion of malignant strictures [48,131]. Perioral cholangioscopy represents an advance in biliary endoscopy that allows a direct endoscopic visualization of the biliary stricture and targeted biopsies. The beneficial role of perioral cholangioscopy was investigated in a systematic review and meta-analysis. The single-operator cholangioscopy (SOC) was observed to be the most accurate diagnostic modality at 96% (95% CI 94–97%), but reaching only a modest pool sensitivity of 65% (95% CI, 35–87%) [132]. However a more recent analysis from de Vries et al., including only selected patients with PSC diagnosis, showed that diagnostic accuracy of SOC was inferior to brushing cytology, with a scarce impact in the diagnostic work-up of those patients [133]. Moreover, most of the analysis on complementary techniques to brushing cytology were conducted regardless of PSC diagnosis. The advantage in combining some of these techniques is lost in PSC patients, with a comparable sensitivity between cytology + FISH alone and cytology + FISH + SOC + transpapillary forceps biopsy [134]. The role of SOC still needs to be evaluated in PSC to select patients that could benefit from the use of this technique.

The use of Endoscopic US (EUS) has been explored, considering its use as a possible alternative to ERCP in the diagnostic work up of biliary strictures in PSC. EUS represents a safer option compared to ERCP [135] and could achieve good diagnostic performances, above all when associated with fine needle aspiration (EUS-FNA), reaching a sensitivity and specificity of 84% and 100%, respectively [136]. However a prior diagnosis of PSC was seen to reduce the diagnostic accuracy of EUS (OR 0.16, 95% CI 0.02–0.99), limiting the enthusiasm about the former results [137]. Moreover, the use of biliary sampling during EUS was associated with a higher risk of tumor seeding [138]. For this reason, undergoing EUS-FNA represents an exclusion criterion for LT protocol from Mayo Clinic for pCCA [139]. Therefore, those patients who are potential candidates for LT should not undergo EUS-guided FNA. Intraductal ultrasound (IDUS), usually complementary to ERCP, gives the possibility of a real-time visualization and characterization of biliary strictures. IDUS has a high sensitivity, even better than ERCP and EUS [140,141], and provides staging information due to the possibility of visualizing the bile duct wall and the layers affected by the neoplasm [142]. However, its role is still limited, and it is infrequently used for the evaluation of suspected biliary strictures. Other advanced techniques, such as probe-based confocal laser endomicroscopy, optical coherence tomography (OCT), or volumetric laser endomicroscopy (VLE), still need more validations to assess their diagnostic accuracy, but early reports are encouraging [135,143,144,145].

## 7. Surveillance of CCA in PSC

The silent development of this neoplasia, typically asymptomatic in early stages, is in contrast with its aggressive behavior and rapid progression. This characteristic makes it difficult to achieve early diagnosis, losing the therapeutical window for curative treatment and compromising patient outcomes, due to limited systemic treatment options. In PSC patients, the radiological surveillance has a crucial role to detect early CCA and to allow curative treatments, such as liver resection or transplantation in patients who are usually young and fit for surgery. An ideal surveillance strategy should be simple, cheap, accepted by the patient, non-invasive, and with high sensitivity, aiming to increase overall survival. However, no diagnostic tests have those characteristics [33], and high-grade dysplasia or early-stage CCA are usually diagnosed using invasive techniques like ERCP, keeping in mind that the risk of complications in ERCP is not justified by the sensitivity of the endoluminal sampling, that is usually low [123]. The positive impact of radiological surveillance for PSC patients has already been proven by different studies, reporting better outcomes with a reduction in hepatobiliary cancers-related deaths in patients exposed to at least a routinary imaging technique [21,146]. Also, the large multi-center study from the IPSCSG proved that scheduled imaging was associated with improved overall survival [33]. Magnetic resonance imaging (MRI) with magnetic resonance cholangiopancreatography (MRCP) is the preferred radiologic modality with no radiation and the possibility to supply more detailed and accurate information than abdominal (US) [25]. The guidelines suggested yearly MRI/MRCP for CCA surveillance in large-duct PSC patients. Pediatric and sdPSC patients represent low-risk categories, and surveillance is not strictly indicated for them [47,98].

There is not a consensus on the correct timing of scheduled imaging for CCA surveillance in PSC, and some experts suggested a limited role of imaging surveillance for biliary malignancies [147]. Recently, a prospective study from Villard et al., including 512 unselected patients followed for 5 years with annual MRI/MRCP, detected CCA in 2% of patients, associated with previous development of progressive and severe biliary strictures. Although 62% of them were eligible for curative treatment, all of them except one developed tumor recurrence and died during the follow-up, pointing out the inability of annual surveillance to detect cancer early enough to provide long-term survival [32]. Moreover, the failing of the guidelines’ strategy has been evident in the group of patients with severe strictures, suggesting how the future direction of research should be the individualization of surveillance, reserving closer MRI/MRCP controls in selected high-risk groups (recent or long-date diagnosis of large-duct PSC, older age, or severe and symptomatic disease).

In addition, as we discussed above, the standardization of both MRI/MRCP sequences acquisition and the radiological report might improve the surveillance efficacy [102,103].

Even without strong evidence, we schedule an MRI/MRCP with DWI sequences every 6 months in PSC patients (every 12 months with contrast enhancement, or before according to clinical indications) in order to detect CCA development as early as possible. However, more prospective multicenter data will be needed to confirm this policy, also from a cost-effective issue. The possible future scenario could be to define different time-intervals for radiological surveillance according to clinical and molecular risk factors in specific patient populations.

The use of scheduled ERCP associated with MRI/MRCP does not improve the surveillance efficacy [33,48], and PSC itself represents a risk factor for ERCP-associated complications [148]. Scheduled ERCP in PSC could have a beneficial effect in the treatment of high-grade stenosis, regardless of patient symptoms according to recent data [149], but does not represent an adequate tool for surveillance.

Abdominal US remains the cornerstone for the surveillance of the other types of hepatobiliary neoplasms occurring in PSC; surveillance of GBC requires a 12-month US surveillance, which could be replaced with or added to the annual MRI/MRCP for CCA surveillance if already performed. A 6-month US is mandatory in the presence of cirrhotic evolution for HCC surveillance [47,98].

The majority of PSC international guidelines discouraged the 6-month use of CA19-9 for CCA surveillance [35,47,98,104] because it has a low positive predictive value and a low accuracy [117]. Indeed, CA19-9 is highly variable and many other benign or malignant conditions, as well as interpersonal variability, could influence the serum levels of these markers. However, the rapid increase of CA 19-9 has been shown to predict malignancies regardless of the cut-off in use, and some experts have tried to narrow the interval of assessment of the biomarker (every 2 or 3 months) in order to detect the progressive CA19-9 change rather than focus on its absolute value [107]. The risk of over-diagnosis associated with this strategy has not been evaluated yet, and the potentially higher rate of false positives could remarkably affect the costs of the surveillance program.

## 8. Future Perspective: Next-Generation Biomarkers for PSC-CCA Diagnosis and Surveillance

Many biomarkers had been tested during the year to find new non-invasive diagnostic strategies (Table 1). The pancreatic autoantibody against glycoprotein 2 (anti-GP2 IgA) had been indicated as a novel marker in large-duct PSC. It represents a prognostic marker associated with a more severe disease and dismal prognosis with a higher risk of developing CCA [150]. Another observation on biochemical markers reveals that a high enhanced liver fibrosis (ELF) score correlates with patients developing CCA sporadically or on PSC. A following multivariate analysis also proved that ELF score was correlated to CCA independently of liver disease stage or patients’ age, suggesting this marker to be related to the biological nature of tumor development [151]. This follows that some markers could be a useful tool for patient risk stratification, being able to identify patients with a higher risk of developing PSC-CCA. This could guide to a more individualized surveillance strategy in PSC. A recent study from Cuenco et al. analyzed the diagnostic performance of a panel of new biomarkers including pyruvate kinase M2 (PKM2), cytokeratin 19 fragment (CYFRA21.1) and mucin 5AC (MUC5AC) achieving promising results with an high specificity (90%) but mostly a high sensitivity when combined with gamma-glutamyltransferase (GGT) (82%) [152]. Also, osteopontin, usually upregulated in CCA tumor cells, was seen to be higher in the serum of patients with CCA compared to healthy controls. During the exploratory analysis of the study from Loosen et al., some 10 PSC patients and 13 healthy controls were compared with 27 patients with CCA, revealing significantly elevated osteopontin levels in CCA patients compared to serum samples from healthy and PSC patients, suggesting a possible role in differentiating among PSC patients those who develop biliary malignancy [153]. Unfortunately, no PSC patients were included in the final analysis, and the observation remains limited to the small exploratory cohort.

A study on the use of volatile organic compounds (VOCs) measurement in bile showed interesting results, revealing differences in gas concentrations between PSC and PSC-CCA patients [154], as already observed in other types of cancer [155]. The subsequent study from the same group of Navaneethan et al. also showed accuracy of VOCs in distinguishing CCA from PSC patients when measured in urine. Some VOCs like ethane and 1-octene reached an 80% sensitivity at 100% specificity, and also 2-propanol and acetonitrile showed promising results, representing a potential non-invasive marker. In the future, the possibility of translating VOCs measurement to breath analysis might be tested [156].

DNA hypermethylation of gene promoters represents an early event in biliary carcinogenesis. The methylation of CDKN2A (p16) represents a frequent DNA alteration [157], but also other loci were found to be hypermethylated in CCA samples [158,159,160], representing potential markers for PSC-CCA detection. However, only a few studies included PSC patients, and data on PSC-CCA are limited. The analysis of methylation status of some candidate genes (cysteine dioxygenase type 1 (CDO1), cannabinoid receptor interacting protein 1 (CNRIP1), septin 9 (SEPT9), and vimentin (VIM)) on biliary brushing of CCA and PSC patients revealed good performances in discriminating malignancy from the benign and inflammatory-related biliary changes in PSC, with high sensitivity (85%) and specificity (98%) [161]. A more recent study tried to validate the measurement of methylation level of the same gene panel in bile, using droplet digital PCR (ddPCR) for the analysis [162]. Previous studies on bile samples could not find a significative association between hypermethylation of specifical genes (CDKN2A) and PSC-CCA [163], or measured potential biomarkers only in non-PSC patients (like CCND2, CDH13, GRIN2B, RUNX3, and TWIST) [164]. Instead, the three center-based (Sweden, Norway, Finland) study from Vedelet et al. observed the gene methylation panel including CDO1, CNRIP1, SEPT9, and VIM to be accurate for early PSC-CCA detection, with an AUC of 0.88. The results were even enhanced including only patients with early PSC-CCA diagnosis (≤12 months) and PSC controls with long follow-up (>36 months), reaching a sensitivity and specificity of 100% and 93%, respectively, and an AUC of 0.98. This promising result proved that DNA methylation biomarkers in bile has the potential to complement standard diagnostic modalities for early PSC-CCA detection [162]. Prospective validation of this study in larger multicenter sample series is warranted. However, the possibility to include this methylation analysis in CCA surveillance in PSC patients remains controversial due to the invasive nature of bile sampling. A few studies also performed DNA methylation analysis on blood samples, discovering some potential biomarkers, already tested in tissue samples [165] (SHOX2, SEPT9, OPCML, HOXD9, etc.), but unfortunately no PSC patients had been included [166,167]. The analysis of micro ribonucleic acid (miRNAs) was highly investigated. miRNAs represent non-coding small nucleic acids capable of binding to specifical messenger RNA (mRNAs) and regulating gene expression [168,169]. Several miRNAs share an important role in chronic cholestasis, interfering with several cellular processes including proliferation and apoptosis. Impaired levels of miRNAs could result in dysregulating cellular cycle generating clonal expansion and eventually cancer. Interestingly, miRNAs expression is not directly altered by cholestasis; indeed, presence of high levels of bilirubin does not influence the result of miRNAs analysis [93]. As already explained above, bile levels of some miRNAs (miR-640, miR-3189, miR-1537, and miR-412) are significantly higher in PSC-CCA patients compared with PSC-alone patients, representing a potential diagnostic tool in early diagnosis. Another study from Li et al. aimed to investigate potential biomarkers for CCA diagnosis regardless of PSC, showing the diagnostic value of an miRNA panel specifically associated with CCA with good sensitivity of 67% and high specificity of 96%. The study included a few PSC patients in the control group, suggesting their possible role in differentiating the miRNA pattern [170]. This aspect needs to be considered in future studies. Some differences were found between miRNAs levels of PSC and CCA-alone patients (miR-1281, miR-126, miR-26a, miR-30b, and miR-122). Nevertheless, a consistent overlapping expression of most miRNAs in PSC and CCA patients was observed. Only miR-126 gave considering results, but only reached a specificity of 93% with poor sensitivity, comparable with brushing cytology [93]. However, another study of miRNA in serum showed a difference in the expression of miR-222 and miR-483-5p in CCA versus PSC controls [171]. Other miRNAs (miR-21, miR-221, miR-122, miR-192, miR-29b, and miR-155) were demonstrated to be associated with CCA in terms of both diagnosis and prognosis; however, control groups of these studies count only a few or no PSC patients [172,173]. Future research is required to investigate on serum miRNAs detection, being less invasive compared to bile sampling. Combined strategies should be implemented to include more information from both serum and bile and integrate them with other biomarkers.

Neoplastic cells are characterized by high metabolism with production of several potential biomarkers. Indeed, metabolic alteration could reflect on impaired concentration of circulating metabolites in the presence of cancer. Specific alteration at the metabolome analysis had been already described in other neoplastic diseases [174,175]. According to that, the group of Banales et al. performed a metabolomic analysis of 424 different metabolites in serum to identify a significant concentration of certain circulating metabolites in patients who developed hepatic malignancy, in particular iCCA or hepatocellular carcinoma (HCC) compared with PSC. Some metabolites were significantly different in the PSC and iCCA group and the algorithms combining some of them, histidine and PC(34:3), could accurately differentiate between the two diagnoses [176]. This suggests a different metabolic profile of CCA and PSC that could represent a starting point for further analysis on PSC-CCA patients.

Promising results also came from proteomic analysis. In particular, urine proteomic analysis reveals the capacity to differentiate CCA from PSC [177]. The use of urine proteomic analysis showed better results when associated with biliary proteomic analysis, which also provides good discrimination capability in the field [178]. The combination of bile and urine proteomic analysis had been explored in a retrospective study and then applied to a prospective analysis, improving sensitivity of CCA diagnosis, compared with standard diagnostic approach, from 72% to 94%. A large prospective trial on patients with progressive cholestasis during surveillance is required to confirm the accuracy of this method for the early diagnosis of CCA [179].

In serum, a glycomic and proteomic analysis on a particular set of proteins was able to identify some differences in the expression of certain glycans in patients with CCA. Among them, fucosylated fetuin A (fc-fetuin A) was capable of differentiating CCA from those with PSC, suggesting a possible role of this analysis in the surveillance of PSC patients [180].

An early field of liquid biopsy-based research is the analysis of extracellular vesicles (EVs) and their proteomic profile in serum samples. EVs contain biomolecules of various nature, like lipids, proteins, and nucleic acids, and different cells, including malignant cholangiocytes, release EVs with specific possible biomarkers [181,182]. First evidence of specifical EVs proteomic patterns came from the analysis of tissue samples, defining several CCA-derived EVs, not in common with PSC, that may reflect tumor features and give prognostic information [183]. Lapitz et al. performed an analysis of EVs’ transcriptomic profile in urine and serum of CCA, showing a specific expression of messenger RNAs (mRNA) and non-coding RNAs (ncRNAs) in CCA patients compared to a mixed group of patients with PSC, UC, and healthy controls. In urine, in the partial analysis including only CCA (*n* = 23) vs. PSC (*n* = 5) patients, a higher expression of some specific mRNAs like CLIP3, VCAM1, and TRIM33 in CCA group (AUC 0.965) was reported, as well as a significant different expression of other transcripts like ATP5EP2, LOC100134713, and SNORA8 [184]. The same analysis in serum (CCA *n* = 12 vs. PSC *n* = 6) reveals mRNAs, PON1, ATF4, and PHGDH were the best candidate biomarkers for the differential diagnosis of CCA and PSC (AUCs 1.00), as well as a high accuracy for the identification of CCA vs. PSC of the transcripts MALAT1, LOC100190986, and SNORA11B (AUCs: 1.00) [184]. Subsequent gene ontology analysis indicated that most of the commonly altered mRNAs participate in carcinogenic pathways. Among EVs biomarkers, proteins also have a differential expression in CCA or PSC (FIBG, A1AG1, S10A8) according recent studies, and interestingly some of them have higher diagnostic values in early-stage CCA (I-II) (FCN2, ITIH4 and FIBG) [183]. A recent proteomic analysis of EVs in serum from the same study group individuates the combination of serum biomarkers CRP/FIBRINOGEN/FRIL as a valid tool for diagnosis of early CCA developing in PSC patients. Other biomarkers were associated with a prognostic role, predicting overall survival. Moreover, the integration of serum PIGR with CRP/FIBRINOGEN/FRIL showed the capacity for predicting CCA development in PSC patients with sensitivity and specificity of 64% and 93%, respectively (AUC 0.91), suggesting its potential use in surveillance settings in the future [185]. Anyway, further studies are needed to validate their effective role and the potential therapeutic strategies associated with their measurement.

Next-generation sequencing (NGS) provides a multigene analysis with high analytical sensitivity and recently grew up with the advances in the characterization of the mutational profile of biliary tumors. Thanks to NGS, many recurrent mutations had been already identified in CCA [186], and some studies already tried to apply NGS for the research of novel biomarkers in de novo CCA diagnosis. A prospective study from Singhi et al. evaluated the use of NGS (a 28-gene panel renamed BiliSeq) in bile duct brushing and tissue sample obtained during ERCP. There was an improvement of sensitivity compared to standard clinical and pathological evaluation (73% vs. 48%). Notably, sensitivity of the BiliSeq panel was considerably higher (83%) in patients with underlying diagnosis of PSC [187]. More recently Arechederra et al. evaluated the use of NGS for the mutational analysis of bile cell-free DNA (cfDNA). The panel in use, called the “Bilemut assay”, achieved a sensitivity of 96.4% and a specificity of 69.2% [188]. The use of bile cfDNA could provide the advantage of obtaining nucleic acids released from all the CCA cells along the biliary tract, overcoming the limitation of the tissue sample and improving sensitivity. In general, the use of NGS for the identification of biomarkers could be useful not only for the early detection of malignancy but also for its prognostic role, being able to detect mutations that could become potential targets of oncological therapies. Further research will fill the gap made by the absence of specifical studies on PSC-CCA patients, testing the diagnostic and prognostic performances of NGS biomarkers panels in this group of high-risk patients.

**Table 1 cancers-15-04947-t001:** Recently investigated biomarkers for PSC-CCA differential diagnosis.

Marker	Sample	Patients Cohort	SE (%)	SP (%)	AUC	Ref.
**Serum biomarkers panel**						
PKM2	serum	CCA (*n* = 66) vs. PSC (*n* = 62)	82%	90%	0.90	[152]
CYFRA21.1						
MUC5AC						
GGT						
ELF score	serum	CCA (*n* = 36) vs. PSC-CCA (*n* = 32) vs. PSC (*n* = 119)	81%	60%	0.74	[151]
**Volatile organic compounds**						
Acrylonitrile + 3-methyl hexane + benzene	bile	PSC-CCA (*n* = 11) vs. PSC (*n* = 21)	91%	73%	0.89	[154]
Ethane + 1-octene	bile	PSC-CCA (*n* = 11) vs. PSC (*n* = 21)	80%	100%	0.90	[154]
2-propanol + Acetonitrile	urine	CCA (*n* = 6) vs. PSC (*n* = 10)	83%	85%	0.86	[156]
2-propanol + carbon disulfide + trimethyl amine	urine	CCA (*n* = 6) vs. PSC (*n* = 10) vs. benign stenoses (*n* = 29)	93%	62%	0.83	[156]
**DNA Methylation markers**						
CDO1 CNRIP1 SEPT9 VIM	biliary brushing	CCA (*n* = 34) PSC (*n* = 34)	85%	98%	0.94	[161]



CDO1	bile	CCA-PSC (*n* = 38) vs. PSC (*n* = 205)	79%	90%	0.88	[162]
CNRIP1						
SEPT9		CCA-PSC ≤ 12 months (*n* = 28) vs. PSC (*n* = 205)	100%	90%	0.98	[162]
VIM						
		CCA-PSC ≤ 12 months (*n* = 28) vs. PSC > 36 months (*n* = 170)	100%	93%	0.98	[162]
**miRNA**						
miRNA191	bile	CCA (*n* = 46) vs. PSC (*n* = 13) vs. benign stenosis (*n* = 37)	67%	96%	-	[170]
U486-3p						
U1274b						
U16						
U484						
miR222	serum	CCA (*n* = 40) vs. PSC (*n* = 40)	-	-	0.77	[171]
miR-483-5p						
miR122	serum	CCA (*n* = 31) vs. PSC (*n* = 40)	32%	90%	0.65	[93]
miR-26a			52%	93%	0.78	
miR-1281			55%	90%	0.83	
miR-126			68%	93%	0.87	
miR30b			52%	88%	0.78	
miR-640	bile	CCA (*n* = 19) vs. PSC-CCA (*n* = 12) vs. PSC (*n* = 52)	50%	92%	0.81	[93]
miR-3189			67%	89%	0.80	
miR-1537			67%	90%	0.78	
miR-412			50%	89%	0.81	
**Metabolomic analysis**						
histidine + PC(34:3)	serum	iCCA (*n* = 20) vs. PSC (*n* = 20)	100%	70%	0.99	[176]
**Proteomic analysis**						
22-peptides CC model	bile	CCA (*n* = 25) vs. PSC (*n* = 18)	84%	78%	0.87	[178]
42-peptides panel	urine	CCA (*n* = 42) vs. PSC/benign stenosis (*n* = 81)	83%	79%	0.87	[177]
Combined BPA/UPA test	urine/bile	CCA (*n* = 16) vs. PSC/benign stenosis (*n* = 29)	94%	76%	0.84	[179]
**Glycomic + proteomic analysis**						
Fucosylated fetuin A	serum	CCA (*n* = 20) vs. PSC (*n* = 39)	62%	90%	0.82	[180]
**EVs mRNA**						
PON1	serum	CCA (*n* = 12) vs. PSC (*n* = 6)	100%	100%	1.00	[184]
ATF4			100%	100%	1.00	
PHGDH			100%	100%	1.00	
CLIP3	urine	CCA (*n* = 23) vs. PSC (*n* = 5)	87%	100%	0.97	[184]
VCAM1			87%	100%	0.97	
TRIM33			87%	100%	0.97	
**EVs non-coding RNA**						
MALAT1	serum	CCA (*n* = 12) vs. PSC (*n* = 6)	100%	100%	1.00	[184]
LOC100190986			100%	100%	1.00	
SNORA11B			100%	100%	1.00	
ATP5EP2	urine	CCA (*n* = 23) vs. PSC (*n* = 5)	87%	100%	0.94	[184]
LOC100134713			83%	100%	0.93	
SNORA8			83%	100%	0.92	
**EVs proteins**						
FIBG	serum	CCA (*n* = 43) vs. PSC (*n* = 30)	88%	63%	0.80	[183]
A1AG1			77%	70%	0.79	
S100A8			70%	67%	0.76	
FCN2	serum	Early stage CCA (*n* = 13) vs. PSC (*n* = 30)	100%	81%	0.96	[183]
ITIH4			92%	81%	0.88	
FIBG			92%	81%	0.88	
CRP	serum	PSC-CCA (*n* = 22) vs. PSC (*n* = 45)	64%	93%	0.91	[185]
FIBRINOGEN						
FRIL						
PIGR						

SE: sensitivity; SP: specificity; AUC: area under the curve.

## 9. Conclusions

Recent scientific advances in PSC-related CCA have improved our knowledge of this disease, both in molecular mechanisms and pathological features, but we are still unsatisfied. The small sample is a major drawback in many studies, and a strengthening of the scientific network is mandatory in this field. The current pathogenetic knowledge of PSC-CCA highlights how it appears to be a different entity from sporadic CCA and explains why it is characterized by a unique clinical presentation and epidemiology. CCA is the most common neoplasm developing in PSC patients associated with high mortality rate, therefore, imaging surveillance is mandatory in these patients. Even without strong evidence, a closer surveillance with MRI/MRCP also including DWI, validated by an expert radiologist, could be a possible strategy for increasing early CCA diagnosis, allowing curative treatments, such as liver resection or LT. In addition, the identification of specific risk factors and possible new algorithms to stratify tumor surveillance based on patient’s individual risk are needed. In parallel, new molecular biomarker identification for the diagnosis of PSC-CCA may have a dual role: screening and diagnosis. Still, integrative genome analysis revealing specific molecular alterations may improve the discovering of future target therapies in PSC-CCA.

## Figures and Tables

**Figure 1 cancers-15-04947-f001:**
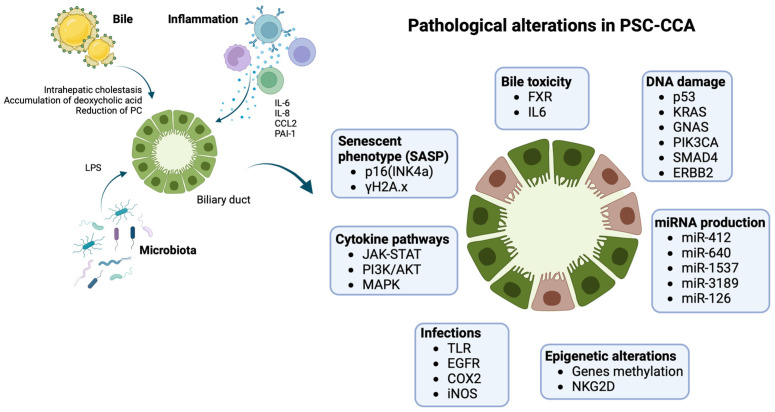
Pathogenesis of PSC-CCA: biliary ducts in PSC are exposed to several pathogenetic triggers, including impaired bile composition, infections, chronic inflammation, ending in damage cholangiocytes on several molecular levels. Inflammation is mediated by different cytokines (IL6, IL8, CCL2) activating pathways such as JAK-STAT, PI3K/AKT, and MAPK, with a proliferative and anti-apoptotic effect on cells. Chronic cholestasis and infections progressively generate up-regulation of iNOS with oxidative stress and stimulation of EGFR and COX2 with increased cellular growth, resistance to apoptosis, and reduction in DNA repair mechanisms. Cholestasis also has transcriptional effects with down-regulation of FXR and reduction of its chemoprotective effect. PSC-CCA is associated with several miRNAs’ dysregulated expression and epigenetic changes that interfere with normal cellular transcription. The effect is the development of an aberrant senescent phenotype. Senescent cholangiocytes are more prone to DNA damage, usually involving oncogenes and oncosuppressors (K-ras, p53), leading to cellular cycle escape and developing of CCA. Images created with BioRender.com, accessed on 18 September 2023.

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
