# Peer review of "Primary Sclerosing Cholangitis-Associated Cholangiocarcinoma: From Pathogenesis to Diagnostic and Surveillance Strategies"

_cancers, 2023, doi:10.3390/cancers15204947_

Round 1
Reviewer 1 Report
Title: Primary sclerosing cholangitis- associated cholangiocarcinoma: from pathogenesis to diagnostic and surveillance strategies
This paper describes review of the major updates in primary sclerosing cholangitis -associated cholangiocarcinoma to highlight the possible future diagnostic and surveillance strategies.
This paper including relatively large number of references, but there are some questions and the author is requested to add the description according to comments as below.
Major points
Decision algorithm for CCA surveillance
There is no detailed decision algorithm for CCA surveillance.
The author should add the decision algorithm for CCA surveillance for further understanding.
Author Response
Dear reviewer,
Thank you for your time on this review. We sincerely appreciated your comment for improving the quality of our work.
Comment 1: Decision algorithm for CCA surveillance. There is no detailed decision algorithm for CCA surveillance. The author should add the decision algorithm for CCA surveillance for further understanding.
Response 1: You have pointed out a crucial issue in the management of PSC patients, therefore we have implemented the related section in the manuscript. Please, find the changes underlined in the re-submitted file.
We hope that our changes will significantly improve the paper. If you have any further suggestions or comments, do not hesitate to contact us.
Best regards,
Elisa Catanzaro on behalf of all authors
Reviewer 2 Report
The most common cause of mortality in people with primary sclerosing cholangitis (PSC) is cholangiocarcinoma (CCA), which is a common malignancy. Risk factors for it include advanced age, male sex, inflammatory bowel disease (IBD), and biliary stenosis. It frequently manifests quickly after PSC diagnosis. In this review, the development of this condition is influenced by a number of molecular pathways, including inflammation and genetic alterations has been described. Early detection is difficult, and the surveillance methods in use today have their limits. Promising new biomarkers for PSC-CCA diagnosis and surveillance include microRNAs, gene methylation, proteomic profiling, and extracellular vesicle components. Better diagnostic and surveillance techniques may result from the integration of these strategies.
I thoroughly enjoyed reading this manuscript and did not find any major issues.
Comments-
1-The gut microbiome may be involved in the onset and progression of PSC, according to some studies. Studies have revealed changes in the diversity and composition of gut bacteria, as well as other abnormalities in the gut microbiome of people with PSC. However, it is unclear exactly how these modifications will affect PSC and how much they will affect it. Therefore also need to discuss it in the text.
2- The author needs to add more information to the graphics, such as details about the molecular mechanism and the relationship between the PSC and CCA.
3-Please specify the software that was used to create this image.
Minor correction needed
Author Response
Dear reviewer,
Thank you for your time on this review. We sincerely appreciated your comments for improving the quality of our work.
Comment 1: The gut microbiome may be involved in the onset and progression of PSC, according to some studies. Studies have revealed changes in the diversity and composition of gut bacteria, as well as other abnormalities in the gut microbiome of people with PSC. However, it is unclear exactly how these modifications will affect PSC and how much they will affect it. Therefore also need to discuss it in the text.
Response 1: The potential involvement of the gut microbiome in the onset and progression of PSC is a crucial aspect of the disease, however the role of the documented alterations of both the gut and the biliary tree, remains largely unknown. To date, some reported pathogenic mechanisms were associated with the development of PSC itself, while more data will be needed to correlate microbiota changes and the development of PSC-CCA. Anyhow, we have added in the section titled "4. Pathogenesis" a list of potential mechanisms of biliary damage in both PSC and PSC-IBD, followed by a brief overview of principal microbiota alterations in PSC. We have also included a reference to relevant studies on the microbiota and CCA. Please, find the changes underlined in the re-submitted file.
Comment 2: The author needs to add more information to the graphics, such as details about the molecular mechanism and the relationship between the PSC and CCA
Response 2: We tried to create a comprehensive figure representing all molecular mechanisms involved in PSC-CCA, however it would have been too complex and not clear for the readers. In the Figure 1, we added a list of principal molecular effectors mechanisms which are more detailly described in the manuscript.
Comment 3: Please specify the software that was used to create this image
Response 3: The software used for the Figure 1 is called Biorender, as previously specified after Publisher’s Note in the text. We added this information below the figure.
We hope that our changes will significantly improve the paper. If you have any further suggestions or comments, do not hesitate to contact us
Best regards,
Elisa Catanzaro on behalf of all authors
Reviewer 3 Report
I have read with interest the review by Catanzaro E et al. The work is comprehensive, clear, and provides an overview of a specific and somewhat overlooked topic. In relation to the objectives of discussing pathogenesis, diagnosis, and surveillance, the text is comprehensive and without gaps. The concepts are scientifically accurate and expressed clearly. As a minor recommendation, I would suggest that they add a few lines about any genetic peculiarities of PSC-CCA compared to CCA (for example, a higher incidence of p53 alterations).
Please, I only noticed minor spelling errors. Thank you
Author Response
Dear reviewer,
Thank you for your time on this review. We sincerely appreciated your comments for improving the quality of our work.
Comments 1: I would suggest that they add a few lines about any genetic peculiarities of PSC-CCA compared to CCA (for example, a higher incidence of p53 alterations).
Response 1: Thank you for the interesting comment. You have expanded the paragraph about genetics alterations in PSC-CCA in section "4. pathogenesis", mainly citing the interesting and comprehensive work from Kamp et al. about the main differences in terms of genetic mutations between PSC-CCA and CCA alone.
We hope that our changes will significantly improve the paper. If you have any further suggestions or comments, do not hesitate to contact us.
Best regards,
Elisa Catanzaro on behalf of all authors